# Free-Floating Aggregate and Single-Cell-Initiated Biofilms of *Staphylococcus aureus*

**DOI:** 10.3390/antibiotics10080889

**Published:** 2021-07-21

**Authors:** Tripti Thapa Gupta, Niraj K. Gupta, Peter Burback, Paul Stoodley

**Affiliations:** 1Department of Microbial Infection and Immunity, The Ohio State University, Columbus, OH 43210, USA; niraj.gupta@osumc.edu (N.K.G.); paul.stoodley@osumc.edu (P.S.); 2Department of Biomedical Education and Anatomy, The Ohio State University, Columbus, OH 43210, USA; burback.1@buckeyemail.osu.edu; 3Department of Orthopedics, The Ohio State University, Columbus, OH 43210, USA; 4National Centre for Advanced Tribology at Southampton (nCATS), Mechanical Engineering, University of Southampton, Southampton SO17 1BJ, UK; 5National Biofilm Innovation Centre (NBIC), University of Southampton, Southampton SO17 1BJ, UK

**Keywords:** *Staphylococcus aureus* (SA), orthopedic infections, aggregates, synovial fluid, biofilms

## Abstract

Periprosthetic joint infection (PJI) occurring after artificial joint replacement is a major clinical issue requiring multiple surgeries and antibiotic interventions. *Staphylococcus aureus* is the common bacteria responsible for PJI. Recent in vitro research has shown that staphylococcal strains rapidly form free-floating aggregates in the presence of synovial fluid (SF) with biofilm-like resistance to antimicrobial agents. However, the development of biofilms formed from these aggregates under shear have not been widely investigated. Thus, in this study, we examined the progression of attached biofilms from free-floating aggregates. Biofilms were grown for 24 h in flow cells on titanium discs after inoculation with either pre-aggregated or single planktonic cells. Image analysis showed no significant difference between the biofilm formed from aggregates vs. the planktonic cells in terms of biomass, surface area, and thickness. Regarding antibiotic susceptibility, there were 1 and 2 log reductions in biofilms formed from single cells and aggregates, respectively, when treated with vancomycin for 24 h. Thus, this study demonstrates the formation of biofilm from free-floating aggregates and follows a similar developmental time period and shows similar antibiotic tolerance to more traditionally inoculated in vitro flow cell biofilms.

## 1. Introduction

Periprosthetic joint infection (PJI) is a devastating and challenging complication after total joint arthroplasty [1,2]. Total knee arthroplasty (TKA) is the most commonly performed arthroplasty, which is then followed by total hip arthroplasty (THA), with over 1 million of these combined replacements carried out in the United States annually [3]. There were 332,000 total hip and 719,000 total knee arthroplasties in 2010, and those numbers are expected to reach 572,000 and 3.48 million by 2030 for hips and knees, respectively [4,5], in the United States. Despite infection control methods implemented to prevent PJI, including antibiotic prophylaxis, patient risk stratification, the detection and treatment of *S. aureus* skin colonization, and highly controlled operating room environments, there is a 1% infection rate for total joint arthroplasty (TJA) [6]. Since the number of primary and revision TJAs are rapidly increasing due to the growing size of the aging population, the number of PJI cases is expected to increase proportionally, as is the number of infections, since the infection rate has stubbornly remained stable over the decade [7]. A major complication in diagnosing and treating PJI is due to biofilm growing on the surfaces of the implant and periprosthetic tissue. Bacterial biofilms are aggregates of microorganisms adhered to biotic or abiotic surfaces. Biofilm-like aggregates can also be present suspended in liquids [8,9]. These microorganisms are enclosed in a self-produced extracellular polymeric substance (EPS) that is composed of nucleic acids, proteins, and polysaccharides [10].

*S. aureus* is the prominent bacteria responsible for causing PJI with biofilms and aggregates observed on the surface of implanted joint devices and the surrounding tissue [11]. Bacterial aggregates have been observed in synovial fluid (SF) from PJI patients [12] as well as both wound and lung infections, indicating they have a broad significance in infection [13,14,15,16]. Bacterial aggregates are formed as host factors such as fibrinogen, fibronectin, and collagen form bridging bonds between the bacterial cells. Bay et al. detected Staphylococci aggregates at wound edges, indicating the initiation of biofilm development [13]. Bacteria in these aggregates are enclosed in the extracellular polymeric matrix as found in biofilms, as evidenced by the study [15]. Recent in vitro studies have shown that *staphylococcus* rapidly forms aggregates in human and large animal SF [12,17,18,19]. Dense aggregates of *S. epidermidis* were also found in SF by Kimberly et al. [19]. SF is a viscous substance in the joint space that holds bactericidal activity [20]. It has been shown that these bacterial aggregates in SF are resistant to the antimicrobial treatments and phagocytosis, similar to the biofilms [17,21,22].

We previously looked at the attachment of free-floating aggregates to different surfaces, including titanium (Ti), as it has been shown as a favorable surface for biofilm growth [23,24] and often used in orthopedics and dental implants [25,26,27]. Our study showed a decrease in surface attachment in SF-induced aggregation after 5 and 15 min of seeding cells [28,29]. This reduction appeared to be due to the proteins in the SF (fibronectin and fibrinogen). However, bacterial surface adhesion was partially restored after the degradation of SF proteins [29]. Therefore, this study aimed to examine the bacterial attachment for longer time periods than 15 min, as the attachment rates and subsequent biofilm formation may be different between the aggregates and the single cells over longer time scales.

Thus, in the present study, we investigated and compared the development of biofilm seeded with SF-mediated preformed suspended aggregates and single cells to compare biofilm development. No significant difference was found in the biofilm formed between the aggregates and the single cells at different starting cell concentrations in terms of the biomass, maximum thickness, and surface area of the biofilm. The biofilm grown for a day on a Ti disc was also treated with vancomycin (1.5 µg/mL) to study the antimicrobial efficacy of an antibiotic on those biofilms formed from free-floating aggregates and single cells.

## 2. Materials and Methods

### 2.1. Bacterial Strain and Culture Conditions

Bacterial stocks were maintained at −80 °C in 20% glycerol. A GFP-expressing *S. aureus* strain (AH1726) [30] was stored and cultured in tryptic soy broth (TSB) (Fisher Scientific, Hampton, NH, USA). A single bacterial colony from the streaked agar plate was used to inoculate 25 mL of TSB for each experiment. The culture tubes were incubated at 37 °C overnight (17–18 h) under shaking (Lab-line instruments, Melrose Park, IL, USA) at 200 rpm.

### 2.2. Bacterial Aggregate Formation

One milliliter of the overnight culture of *S. aureus* (10^8^ CFU/mL) was centrifuged at 21,000× *g* for 1 min. The supernatant was removed, and the pellet was washed in PBS and then resuspended in 10% bovine SF (100 µL) in 900 mL of ringer’s solution (RS). The cells were then incubated in static conditions for 1 h to allow aggregate formation at 37 °C. For single cell suspension preparation, 1 mL of the overnight culture was centrifuged at 21,000× *g* for 1 min and after removing the supernatant, the pellet was washed and resuspended in 1 mL of RS. After 1 h, the aggregates and single-cell suspensions were injected through a flow cell (Figure 1) with a 1 mL syringe.

### 2.3. Biofilm Growth Assay

Two different concentrations of 10^8^ CFU/mL and 10^3^ CFU/mL were used for biofilm growth in a flow cell with chamber dimensions of 39 × 13 × 0.34 mm (FC 270-AL, BioSurface Technologies, Bozeman, MT, USA). Ti (grade 2) discs of 10 mm diameter and 2 mm thickness (FC 110-Ti, Biosurface Technologies, Bozeman, MT, USA) were placed in a flow cell for biofilm growth assay. After injecting the respective concentration of aggregates and single-cell suspensions, the flow cell was incubated at 37 °C under static conditions to allow attachment for 1.5 h. Sterile TSB medium was then flowed through the flow cell using a peristaltic pump (IPC ISM932A, Cole-Parmer, Vernon Hills, IL, USA) with a flow rate of 450 µL/min creating a shear stress of 15 mPa and run for 24 h for biofilm growth.

### 2.4. Ti Disc Modification and Characterization

Ti discs were sanded using an aluminum oxide sanding sheet (436A38, Grainger P600, USA) for 4–5 min. The roughness of these discs was measured using a Zeta 20 Optical Profilometer and the Ra value was calculated as published elsewhere [28].

### 2.5. Confocal Laser Scanning Microscopy (CLSM) Assessment of Biofilms Formed from Aggregates and Single Cells on Ti Discs

The biofilm formed on discs were imaged under 60× magnification using an Olympus FluoView (FV10i, PA, USA) Confocal Laser Scanning Microscope (CLSM). The images obtained were analyzed by COMSTAT (comstat2, Denmark) [31,32] which takes biofilm image stacks recorded by a confocal microscope and measures various biofilm structures. For COMSTAT analysis, z-stack images were taken from three discs. From each disc, three random regions were selected, and z-stack images were subsequently taken to measure the biomass, surface area, and maximum thickness of the biofilm. For staining the biofilm components, wheat germ agglutinin alexa fluor 647 (W32466, Fisher Scientific, USA; final concentration, 5 µg/mL), DAPI solution (EN62248, Fisher Scientific, USA; final concentration, 1 µg/mL), and SYPRO ruby biofilm matrix stain (F10318, Fisher Scientific, USA) were used to stain the poly-N-acetylglucosamine (PNAG) or SF-derived hyaluronic acid (HA) residues, extracellular DNA (eDNA), and matrix proteins, respectively.

### 2.6. Antibiotic Treatment Exposure

Biofilms grown for 1 day on Ti disc were taken out from the flow cell and washed twice in PBS. The disc was then immersed in a well containing 1.5 µg/mL vancomycin made in water and incubated at 37 °C for 3 and 24 h. Following incubation, the disc was again washed with PBS thrice and kept in a conical tube with 5 mL of PBS for sonication. Five minute sonication followed by 3 min vortex was carried out to dislodge the biofilm attached on the surfaces. The bacterial suspensions were then plated for bacterial enumeration with the micro dilution method.

### 2.7. Statistical Analysis

All experiments in this study were repeated three times. The threshold for significance was set at *p* < 0.05. Statistical significance was determined by a Student’s *t*-test in Excel. All error bars in the charts indicate the standard error of the means (SEM).

## 3. Results

### 3.1. Biofilms Formed from SF Induced Aggregate and Single Cells on Ti discs

To investigate the biofilm formation between SF-induced aggregates (+SF) and single cells (-SF), confocal microscopy was used to qualitatively evaluate the formation of biofilms (Figure 2 and Figure 3). Two different cell concentrations of 10^8^ CFU/mL and 10^3^ CFU/mL were used to form biofilm. At both higher and lower cell concentrations, biofilm was formed in both +SF and -SF. Furthermore, CLSM analysis evidenced different biofilm components such as eDNA, PNAG, and matrix proteins present in the biofilms. For both +SF and -SF, all the matrix components were efficiently stained with matrix stains, colocalizing with green fluorescent protein (GFP) signal. For the stain wheat germ agglutinin (WGA), it was not possible to discriminate whether the signal was derived from PNAG produced by bacteria or from incorporated hyaluronic or from other N-acetylglucosamine (GlcNAc)-containing polysaccharides resident in the SF [33].

### 3.2. COMSTAT Analysis of Biofilm Formed from SF Induced Aggregates and Single Cells

Z-stacks were collected to generate three-dimensional views for COMSTAT analysis, thereby calculating the biomass, thickness, and surface areas of the biofilm with data accumulated for three separate z-stacks. COMSTAT analysis showed no significant difference between the +SF and −SF biofilms in higher and lower cell concentrations (Figure 4). For the higher cell concentration of 10^8^ CFU/mL, the biomass for both +SF and −SF were similar, 10 µm^3^/µm^2^ (*p* = 0.5383), whereas for a lower cell concentration of 10^3^ CFU/mL, the biomass for +SF (8 µm^3^/µm^2^) was lower than that of -SF (10 µm^3^/µm^2^) (*p* = 0.3655). This difference in biomass was not statistically significant (*p* > 0.05). The maximum thickness of +SF and −SF biofilms were 21 and 20 µm for 10^8^ CFU/mL (*p* = 0.5725). For 10^3^ CFU/mL, the thickness of the biofilm was 20 µm for both +SF and -SF biofilms (*p* = 0.5879). In terms of surface area covered by the biofilm, +SF contained a greater surface area than −SF for 10^8^ CFU/mL. In contrast, for 10^3^ CFU/mL, −SF had a greater biofilm covered area than +SF.

### 3.3. Vancomycin Treatment of Biofilms Formed from SF Induced Aggregate and Single Cells on Ti

The 24 h biofilms were treated with vancomycin (Van) to observe the antimicrobial efficacy of antibiotics on biofilm formed from aggregates and the single cells. Confocal microscopy images revealed the treatment efficacy of vancomycin on these biofilms (Figure 5, Figure 6, Figure 7 and Figure 8) with different biofilm components such as eDNA and PNAG. In the figures, propidium iodide (PI) shows the dead cells after vancomycin treatment. For both +SF and −SF, all the matrix components were efficiently stained with matrix stains colocalizing with GFP signal. The images were taken after 3 (Figure 5 and Figure 7) and 24 h (Figure 6 and Figure 8) of vancomycin treatment. For both cell concentrations, as shown in Figure 9, the bacterial viability decreased by 0.5 log after 3 h of antibiotic treatment for biofilms formed from SF-induced aggregation. However, not much reduction was observed for biofilms formed from single cells after 3 h of treatment. For +SF and −SF biofilms, 2 and 1 log reductions in bacterial cells were observed after 24 h of vancomycin treatment for both cell concentrations. Similarly, for −SF biofilm, 1 log reduction was observed after 24 h of treatment in higher cell concentrations of 10^8^ CFU/mL, as shown in Figure 9.

## 4. Discussion

*S. aureus* forming free-floating aggregates in human and bovine SF and growing to a macroscopic size in 24 h have been reported in many in vitro studies [12,17,18,19,21,34]. The formation of aggregates provides the bacteria with enhanced tolerance to antibiotics and promotes surface colonization and biofilm formation [35,36]. Therefore, we hypothesized that the aggregates that form in SF provide early protection to bacteria entering the surgical site, allowing them time to attach to the implant surface, leading to biofilm formation. The phenotype shown by aggregates is similar to biofilm in terms of antibiotic tolerance, resilience toward immune response, and a stabilized chemical environment [8,9]. Thus, understanding the initiation of biofilm development from aggregates is important to understand their role in chronic infection, as very little is known in this area.

Through our confocal microscopy analysis, we found that aggregates form biofilms similar to the biofilm formed by the single cells, although with COMSTAT analysis, there was no significant difference between the biofilm formed between these two. The confocal images also showed the presence of biofilm components such as eDNA, PNAG or HA, and the matrix proteins in biofilm formed from aggregates and the single cells. This is similar to a study where different matrix components of clinical SF were stained [33]. Furthermore, the biofilm formed from SF-induced aggregates were clearly stained by WGA. This could be explained by a possible ability of *S. aureus* to bind and incorporate HA in the matrix joint fluids, as explained by other groups [33,37]. The confocal images and COMSTAT analysis further revealed no difference in biofilm formation, when higher and lower bacterial cell concentrations were used to form the aggregates and single cells initiated biofilms. This could be potentially due to biofilm accumulation reaching steady state, even with the lower inoculum, within 24 h. The biomass and the biofilm thickness value showed little difference in terms of biofilm formation between the free-floating aggregates and the single cells. We found that there was 24% more biomass in the biofilm formed by single cells at 10^3^ CFU/mL than the biofilm by the preformed aggregates containing same cell concentration. However, for a higher inoculum concentration of 10^8^ CFU/mL, this difference was only 4%. This could be due to the fact that single cells grow more rapidly than the aggregates formed by a low concentration of cells. On the other hand, the rate of growth of aggregates increases as the size of initial aggregates increases [9,38]. Therefore, we found a relatively small difference in the case of aggregates from higher-density cells. Similarly, the biofilm formed by single cells at 10^3^ CFU/mL was 5% thicker than the one formed by the aggregates with same cell concentration, whereas for the higher inoculum concentration (10^8^ CFU/mL), the biofilm formed by single cells was 11% less thick than the biofilm formed with the aggregates. This could be potentially due to the fact that aggregates formed by cells at a lower concentration are significantly smaller than the aggregates formed by cells at higher concentration [39]. Therefore, biofilms formed by larger aggregates attached on the surface could be relatively thicker than the ones formed by the single cells or aggregates formed by lower concentration. Furthermore, vancomycin was used to observe the antimicrobial efficacy on the biofilm formed from aggregates and the single cells. The vancomycin concentration used was 2XMIC (0.75 µg/mL) of bacteria used in this study. With vancomycin treatment, the bacterial load was reduced by 1 and 2 logs in the biofilm formed from aggregates in both lower and higher cell concentrations, whereas after 24 h vancomycin treatment, only 1 log reduction was observed on single-cell biofilm. Confocal microscopy also shows more dead cells stained by PI in the aggregate’s biofilm than to the single-cell biofilm. This decrease in bacterial viability between the aggregates biofilm and single cells might be because of SF possessing the antibacterial properties on most common Gram-positive and Gram-negative organisms [20]. One group hypothesized that the bactericidal permeability protein from SF might be responsible for its antimicrobial properties due to the cytotoxic nature of this protein to Gram-negative bacteria because of its affinity for the lipopolysaccharide bacterial wall [40]. In the case of Gram-positive bacteria, the proteins such as lysozyme, lactoferrin, and secretory phospholipase might play a role in the bactericidal activity of SF observed against the *Staphylococcus* and *Streptococcus* species, as speculated by other study [20]. These proteins are believed to exist in the SF of healthy, traumatized, or osteoarthritis knees and provide innate defense against Gram-positive organisms [20].

Furthermore, we acknowledge some limitations of this study, such as the characterization of biofilm after 24 h of growth period when the biofilms were relatively saturated. We believe that initial events in terms of bacterial attachment initiated with lower and higher concentration inocula would provide more insights towards the biofilm formation process. Thus, we aspire to conduct this study with shorter time periods to observe the initial events in the biofilm formation process. In addition, we also intend to see the effects of SF-mediated aggregates in the strength of biofilm formed over a longer period of 5–7 days. Therefore, with this study, we presented evidence of the role of aggregates in biofilm formation, and we intend to carry out further research to characterize the early events in the biofilm formation process and the long-term health of mature biofilm that were initiated with SF-mediated aggregates. This series of studies could provide new knowledge based on understanding the biofilm formation process from initiation to maturation in PJIs. In the future, we also intend to use reference and patient-isolated bacterial strains to further explore this study, as the usage of a single bacterial strain might be a limiting factor for conclusive results.

## 5. Conclusions

The obtained results provided evidence that *S. aureus* aggregated in presence of synovial fluid can form biofilm. The formation of biofilm thus follows a similar developmental time period and antibiotic tolerance as more traditionally inoculated in vitro flow cell biofilms. While we acknowledge that future studies will be necessary to fully understand the extent of synovial fluid-initiated biofilm formation, our study provides evidence that *S. aureus* interaction with synovial fluid has an important role during biofilm formation and PJIs.

## Figures and Tables

**Figure 1 antibiotics-10-00889-f001:**
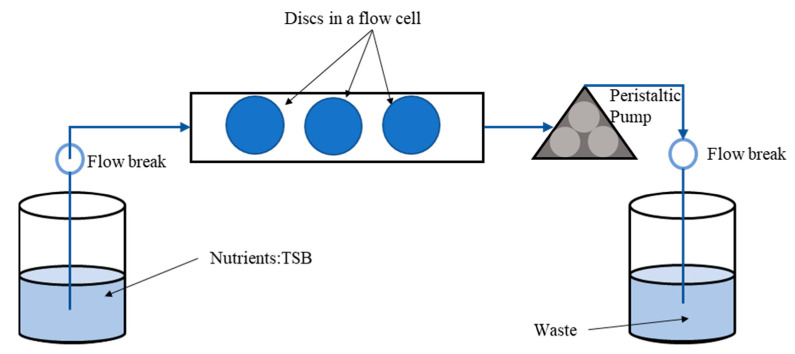
Schematics of experimental setup for growing biofilm in a flow through system.

**Figure 2 antibiotics-10-00889-f002:**
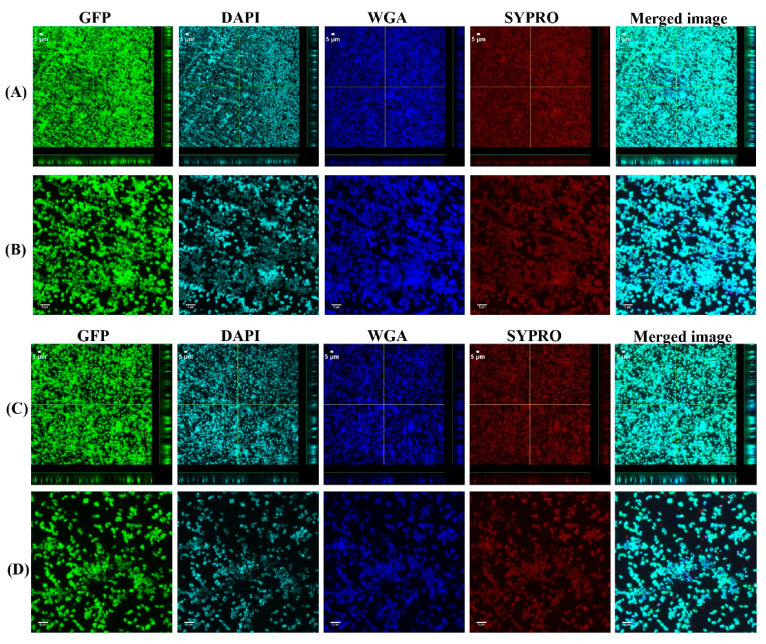
Confocal images of aggregated (**A**,**B**) and single cells (**C**,**D**) biofilms for 10^8^ CFU/mL. Panel B and D are zoomed images. Biofilms stained with DAPI for eDNA, WGA stain for PNAG or SF-derived HA, and SYPRO for staining proteins contained in the biofilm matrix. Merged images denote the combination of GFP, DAPI, WGA, and SYPRO images.

**Figure 3 antibiotics-10-00889-f003:**
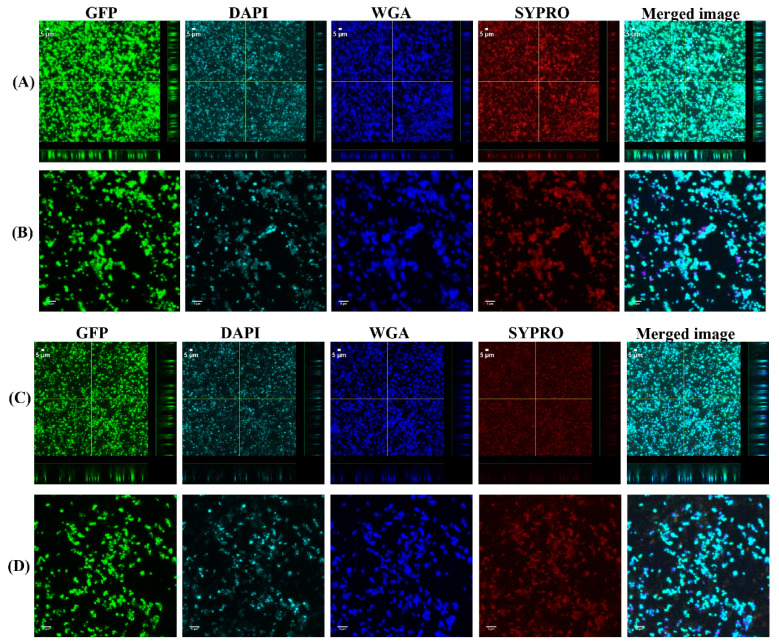
Confocal images of aggregated (**A**,**B**) and single cells (**C**,**D**) biofilms for 10^3^ CFU/mL. Panel B and D are zoomed images. Biofilms stained with DAPI for eDNA, WGA stain for PNAG or SF-derived HA, and SYPRO for staining proteins contained in the biofilm matrix. Merged images denote the combination of GFP, DAPI, WGA, and SYPRO images.

**Figure 4 antibiotics-10-00889-f004:**
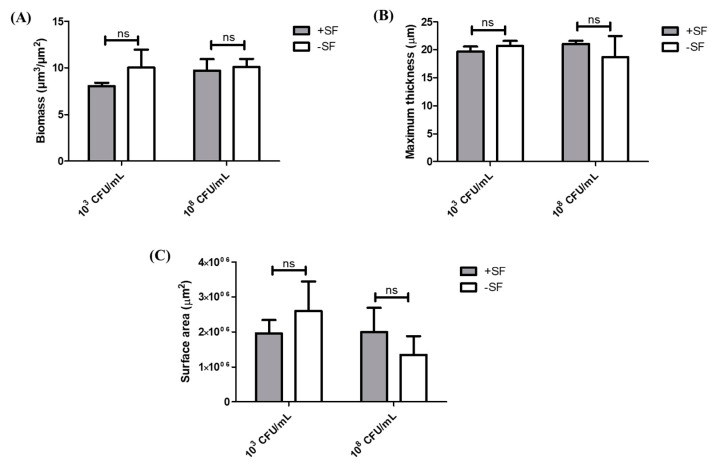
COMSTAT analysis—(**A**) biomass, (**B**) thickness, and (**C**) surface areas. ns = not significant.

**Figure 5 antibiotics-10-00889-f005:**
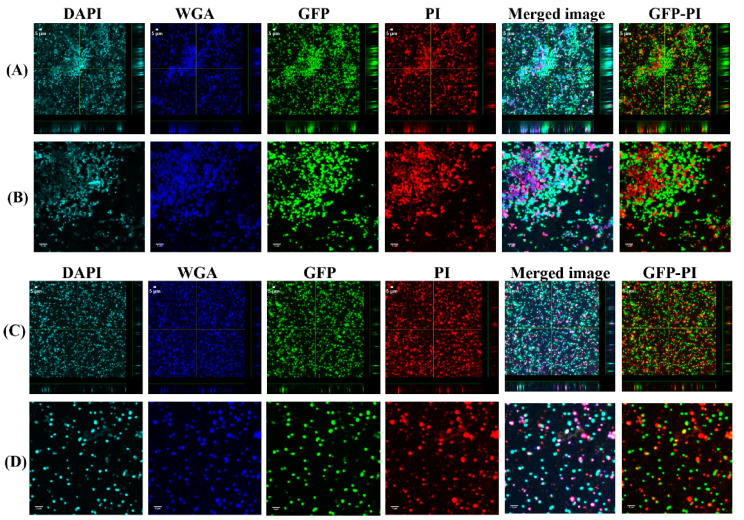
Confocal images of aggregated (**A**,**B**) and single cells (**C**,**D**) biofilms for 10^8^ CFU/mL after 3 h antibiotic treatment. Panel B and D are zoomed images. Biofilms stained with DAPI for eDNA, WGA stain for PNAG or SF-derived HA, and PI for staining dead cells. Merged images denote the combination of DAPI, WGA, GFP, and PI images. GFP-PI is combination of GFP and PI images.

**Figure 6 antibiotics-10-00889-f006:**
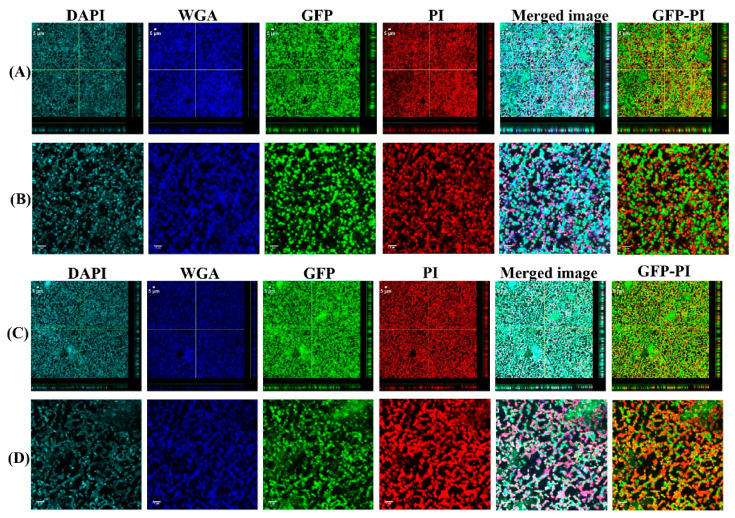
Confocal images of aggregated (**A**,**B**) and single cells (**C**,**D**) biofilms for 10^8^ CFU/mL after 24 h antibiotic treatment. Panel B and D are zoomed images. Biofilms stained with DAPI for eDNA, WGA stain for PNAG or SF-derived HA, and PI for staining dead cells. Merged images denote the combination of DAPI, WGA, GFP, and PI images. GFP-PI is combination of GFP and PI images.

**Figure 7 antibiotics-10-00889-f007:**
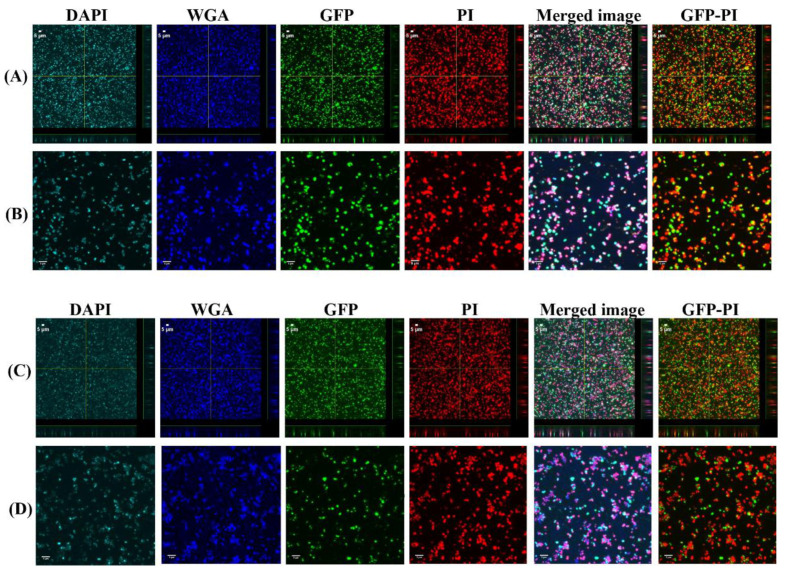
Confocal images of aggregated (**A**,**B**) and single cells (**C**,**D**) biofilms for 10^3^ CFU/mL after 3 h antibiotic treatment. Panel B and D are zoomed images. Biofilms stained with DAPI for eDNA, WGA stain for PNAG or SF-derived HA, and PI for staining dead cells. Merged images denote the combination of DAPI, WGA, GFP, and PI images. GFP-PI is combination of GFP and PI images.

**Figure 8 antibiotics-10-00889-f008:**
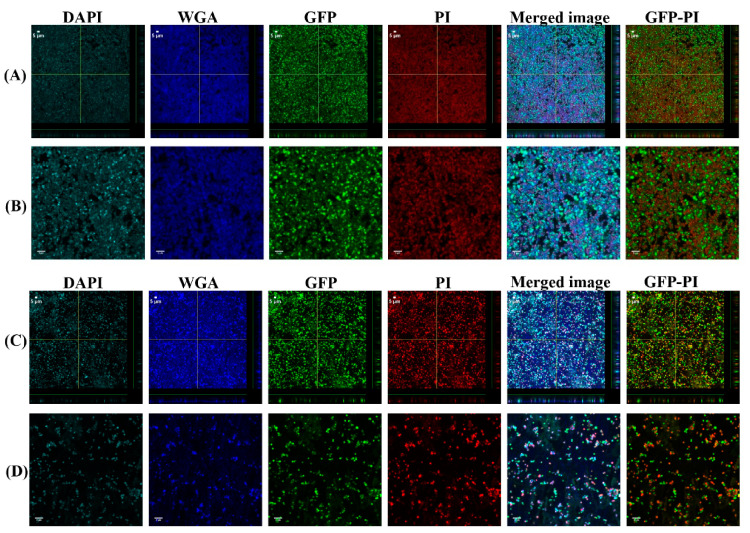
Confocal images of aggregated (**A**,**B**) and single cells (**C**,**D**) biofilms for 10^3^ CFU/mL after 24 h antibiotic treatment. Panel B and D are zoomed images. Biofilms stained with DAPI for eDNA, WGA stain for PNAG or SF-derived HA, and PI for staining dead cells. Merged images denote the combination of DAPI, WGA, GFP, and PI images. GFP-PI is combination of GFP and PI images.

**Figure 9 antibiotics-10-00889-f009:**
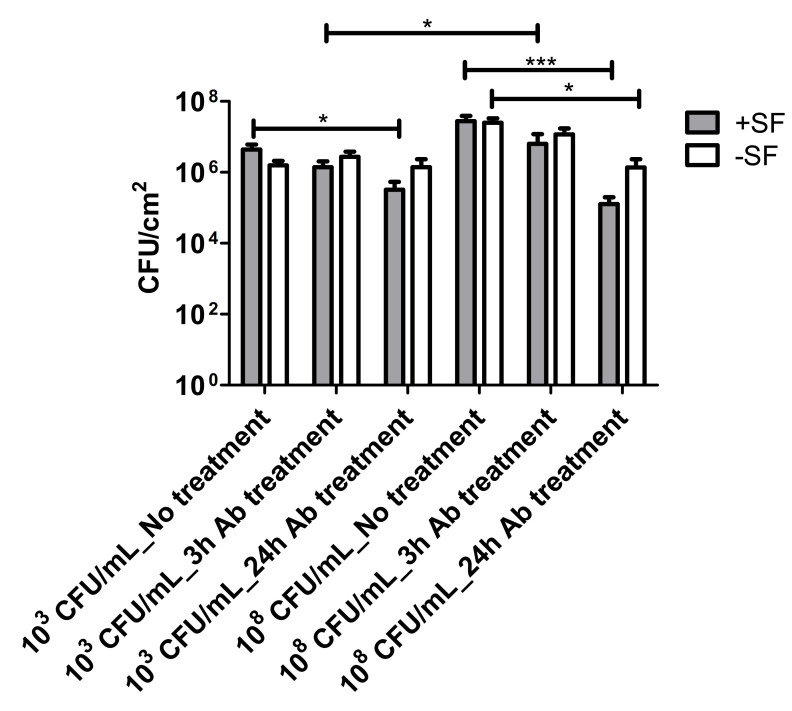
Bacterial viability estimation based on the measurement of CFU after antibiotic treatment for 3 and 24 h. Three replicas of each treatment were performed for each +SF and −SF. * (*p* < 0.05), *** (*p* < 0.001).

## Data Availability

The data presented in this study are contained within the article.

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
