# Peer review of "Free-Floating Aggregate and Single-Cell-Initiated Biofilms of Staphylococcus aureus"

_antibiotics, 2021, doi:10.3390/antibiotics10080889_

Round 1
Reviewer 1 Report
General comment:
The present manuscript is interesting and new as a study! Through the work, I have found some errors in the English used and needs to be reformulated in some places; but after the review, this work can be further evaluated for publication. Please find critical comments as follows:
Critical comments:
Title: I suggest for the title this form: “Free-floating aggregate and single cell-initiated biofilms of Staphylococcus aureus”.
Keywords: I suggest to put the long name of S. aureus “Staphylococcus aureus”, to delete “materials” and to put in plural “biofilms”.
Please don’t repeat so much the same adverb/conjunction next to each other…you need to enrich the paper.
- Introduction
Please improve the English in this section! If you look better, there are some unnecessary sentences/phrases and you need to enrich the Introduction.
Line 36: Put the acronym of S. aureus.
Line 37: Put the acronym of TJA and use it in Line 38.
Line 39: Delete the long name of PJI
Line 46: Put the acronym into brackets for SF.
Lines 53-54: Delete long name of SF and use SF in Line 54; the same in Line 56.
Line 61: Put acronym of SF; the same in Lines 62 & 67.
Lines 66-69: This part is the aim, but it is necessary to be improved.
Lines 69-71: You talk for titanium after the aim, isn’t correct this. Please put it before the aim and I suggest to be referred to this reference that can be helpful for you about the dental implants in vitro: [Differential Efficacy of Two Dental Implant Decontamination Techniques in Reducing Microbial Biofilm and Re-Growth onto Titanium Disks In Vitro. Appl. Sci. 2019, 9, 3191. https://doi.org/10.3390/app9153191].
Lines 71-79: This isn’t part of the Introduction. This looks more as a methodology. You have to delete it and looking for another place to put.
- Materials and Methods
This section has a lot of information and I appreciate this, but also here are some issues to be corrected.
Line 84: Delete the long name of TSB.
Line 86: Here you can add for example a phrase like this: “under shaking overnight…”
Line 90: Put SF.
Lines 95-96: Put a dot at the end of the sentence and delete the last sentence, because it is repeated in the next sub-section (2.3).
Figure 1: I suggest to use the word “discs” and not coupons, because isn’t clear this term. But, if don’t agree with, please find another word to be suitable. Also put the acronym of TSB, you have used the long name in 2.1. Please improve the figure legend here.
Line 100: Correct the verb tense here “were” and not was. In Line 101 put in lower cap “x” for the chambers dimension.
Line 116: Put CLSM into brackets after the long name and add also city and state of the device. I suggest to add some lines into Introduction about the COMSTAT analysis what it is or here adding a little prescription and for the program you have used in your study, please add the version, city, state of it…because is part of the methodology. After you can continue with the rest.
Lines 126 & 129: It isn’t correct to start the new sentence with a number.
Lines 135 & 136: Please add Student t-test and about the standard error of the means
(± SEM), if so.
- Results
Line 138: Put in plural “surfaces”.
Lines 140 & 142: Put in plural: biofilms, concentrations.
Line 143: Improve this sentence adding +SF and -SF. Please, when you use the acronyms for the first time you have to maintain this during the entire manuscript!
Line 146: Put the long names of GFP & WGA.
Line 160: In the title, if you find it reasonable put the acronyms.
Line 161: I suggest to replace the word “renderings” with “views”, if you agree with.
Line 162: Put in plural “areas”.
Figure 4: In the legend put in plural “areas” and also “n.s. = not significant”.
Line 178: Put in plural “Figs”.
Line 179: Put the long name of PI or in the section of Materials and Methods.
Line 185: Put in plural “biofilms”.
Lines 193, 198, 203 & 208: In these legends you wanted to say “of or for”?
Figure 9: In the text or here in the legend put the long name of “Ab” and its acronym into brackets. Delete the long name of CFU and maintain its acronym. Where is the results described for the Figure 9 in the appropriate paragraph? I haven’t found Fig.9 mentioned in the text, but only here.
- Discussion
This section needs improvement in English and rephrasing. You need to mention your newest of this work!
Lines 213 & 217: Put SF.
Line 225: Put semicolon “;” after the COMSTAT analysis, if you don’t want to separate this long sentence and improving the meaning.
Line 231: About the Hyaluronic Acid (HA) in dentistry, I suggest this article:
[Surface Treatment of the Dental Implant with Hyaluronic Acid: An Overview of Recent Data. Int J Environ Res Public Health. 2021 Apr 27;18(9):4670. doi: 10.3390/ijerph18094670.].
Line 232: Put a comma before “when”, because is a long sentence.
Line 238: I think that the correct verb tense here is “were” and please improve the discussion of these two different biofilms.
Line 240: You wanted to say “inoculum” here or not?
Lines 249-251: Put SF.
Lines 254-263: This part is very confused and needs an improvement; the sentences are with two meanings and aren’t all of them so clear to understand your final discussion. You can add something as a future perspective or what is going to help the clinician or…”SO YOUR GOLD MESSAGE WHAT IS IT?”
- Conclusions
This section requires to be clearer; check the English as well!
Lines 265-270: Here you have repeated your results and I suggest to reformulate this paragraph adding your final and real conclusion to be understandable for
Author Response
Responses to reviewer 1 comments
The authors would like to take this opportunity to sincerely thank the reviewers for the time and the constructive comments.
- Reviewer’s comment: The present manuscript is interesting and new as a study! Through the work, I have found some errors in the English used and needs to be reformulated in some places; but after the review, this work can be further evaluated for publication. Please find critical comments as follows:
Authors’ Response: The authors appreciate the comment and the positive feedback.
- Reviewer’s comment: Title: I suggest for the title this form: “Free-floating aggregate and single cell-initiated biofilms of Staphylococcus aureus”.
Authors’ Response: The title has been corrected as per the reviewer’s comment.
- Reviewer’s comment: Keywords: I suggest to put the long name of S. aureus “Staphylococcus aureus”, to delete “materials” and to put in plural “biofilms”.
Authors’ Response: The keywords have been modified as suggested.
- Reviewer’s comment: Please don’t repeat so much the same adverb/conjunction next to each other…you need to enrich the paper.
Authors’ Response: Grammar check has been completed throughout the document and changes were made accordingly.
- Reviewer’s comment: Introduction
Please improve the English in this section! If you look better, there are some unnecessary sentences/phrases and you need to enrich the Introduction.
Authors’ Response: The introduction section has been improved.
- Reviewer’s comment: Line 36: Put the acronym of S. aureus.
Authors’ Response: Acronym of S. aureus has been written.
- Reviewer’s comment: Line 37: Put the acronym of TJA and use it in Line 38.
Authors’ Response: It has been corrected.
- Reviewer’s comment: Line 39: Delete the long name of PJI
Authors’ Response: The long name of PJI has been deleted.
- Reviewer’s comment: Line 46: Put the acronym into brackets for SF.
Lines 53-54: Delete long name of SF and use SF in Line 54; the same in Line 56.
Line 61: Put acronym of SF; the same in Lines 62 & 67.
Authors’ Response: They have been corrected in the manuscript.
- Reviewer’s comment: Lines 66-69: This part is the aim, but it is necessary to be improved.
Authors’ Response: The aim has been improved (line 83-84)- Thus, in the present study we investigated and compared the development of biofilm seeded with SF-mediated preformed aggregates and single cells to compare biofilm development.
- Reviewer’s comment: Lines 69-71: You talk for titanium after the aim, isn’t correct this. Please put it before the aim and I suggest to be referred to this reference that can be helpful for you about the dental implants in vitro: [Differential Efficacy of Two Dental Implant 1.
Decontamination Techniques in Reducing Microbial Biofilm and Re-Growth onto Titanium Disks In Vitro. Appl. Sci. 2019, 9, 3191. https://doi.org/10.3390/app9153191].
Authors’ Response: Titanium has been placed before the aim with suggested citation.
- Reviewer’s comment: Lines 71-79: This isn’t part of the Introduction. This looks more as a methodology. You have to delete it and looking for another place to put.
Authors’ Response: Majority of the sentences have been deleted from this paragraph with addition of few sentences in line 74-76.
- Reviewer’s comment: Materials and Methods
This section has a lot of information and I appreciate this, but also here are some issues to be corrected.
Line 84: Delete the long name of TSB.
Authors’ Response: The long name of TSB has been deleted.
- Reviewer’s comment: Line 86: Here you can add for example a phrase like this: “under shaking overnight…”
Authors’ Response: The phrase “The culture tubes were incubated at 37 °C overnight (17-18 hours) under shaking at 200 rpm” has been added in line 102-103.
- Reviewer’s comment: Line 90: Put SF.
Authors’ Response: Error has been corrected.
- Reviewer’s comment: Lines 95-96: Put a dot at the end of the sentence and delete the last sentence, because it is repeated in the next sub-section (2.3).
Authors’ Response: The last sentence has been deleted.
- Reviewer’s comment: Figure 1: I suggest to use the word “discs” and not coupons, because isn’t clear this term. But, if don’t agree with, please find another word to be suitable. Also put the acronym of TSB, you have used the long name in 2.1. Please improve the figure legend here.
Authors’ Response: The word coupons has been changed to discs and acronym of TSB has been corrected. The figure 1 legend has been changed.
- Reviewer’s comment: Line 100: Correct the verb tense here “were” and not was. In Line 101 put in lower cap “x” for the chambers dimension.
Authors’ Response: The errors have been corrected in the manuscript.
- Reviewer’s comment: Line 116: Put CLSM into brackets after the long name and add also city and state of the device. I suggest to add some lines into Introduction about the COMSTAT analysis what it is or here adding a little prescription and for the program you have used in your study, please add the version, city, state of it…because is part of the methodology. After you can continue with the rest.
Authors’ Response: The errors have been corrected.
Information regarding COMSTAT has been added in the manuscript in line 143-145.
- Reviewer’s comment: Lines 126 & 129: It isn’t correct to start the new sentence with a number.
Authors’ Response: The sentence has been corrected.
- Reviewer’s comment: Lines 135 & 136: Please add Student t-test and about the standard error of the means (± SEM), if so.
Authors’ Response: Suggested changes have been made.
- Reviewer’s comment: Results
Line 138: Put in plural “surfaces”.
Authors’ Response: It has been corrected in the manuscript.
- Reviewer’s comment: Lines 140 & 142: Put in plural: biofilms, concentrations.
Authors’ Response: They have been corrected in the manuscript.
- Reviewer’s comment: Line 143: Improve this sentence adding +SF and -SF. Please, when you use the acronyms for the first time you have to maintain this during the entire manuscript!
Authors’ Response: It has been corrected in the manuscript.
- Reviewer’s comment: Line 146: Put the long names of GFP & WGA.
Authors’ Response: GFP and WGA full names have been added.
- Reviewer’s comment: Line 160: In the title, if you find it reasonable put the acronyms.
Authors’ Response: Although it is reasonable to put the acronyms, the authors prefer the full spelled out title for better readability.
- Reviewer’s comment: Line 161: I suggest to replace the word “renderings” with “views”, if you agree with.
Authors’ Response: Word ‘renderings’ has been changed to ‘views’.
- Reviewer’s comment: Line 162: Put in plural “areas”.
Authors’ Response: It has been corrected.
- Reviewer’s comment: Figure 4: In the legend put in plural “areas” and also “n.s. = not significant”.
Authors’ Response: It has been corrected.
- Reviewer’s comment: Line 178: Put in plural “Figs”.
Authors’ Response: It has been added.
- Reviewer’s comment: Line 179: Put the long name of PI or in the section of Materials and Methods.
Authors’ Response: Long name of PI has been added in the results section.
- Reviewer’s comment: Line 185: Put in plural “biofilms”.
Authors’ Response: It has been corrected.
- Reviewer’s comment: Lines 193, 198, 203 & 208: In these legends you wanted to say “of or for”?
Authors’ Response: ‘HA of SF’ has been replaced with ‘SF-derived HA’ throughout the manuscript.
- Reviewer’s comment: Figure 9: In the text or here in the legend put the long name of “Ab” and its acronym into brackets. Delete the long name of CFU and maintain its acronym. Where is the results described for the Figure 9 in the appropriate paragraph? I haven’t found Fig.9 mentioned in the text, but only here.
Authors’ Response: Figure 9 has been added in the text in line 247. Long name of CFU has been deleted from the figure legend. Long name of Ab has been added in the text (line 240).
- Reviewer’s comment: Discussion
This section needs improvement in English and rephrasing. You need to mention your newest of this work!
Lines 213 & 217: Put SF.
Authors’ Response: SF has been added. Discussion section has been improved.
- Reviewer’s comment: Line 225: Put semicolon “;” after the COMSTAT analysis, if you don’t want to separate this long sentence and improving the meaning.
Authors’ Response: Semicolon has been added.
- Reviewer’s comment: Line 231: About the Hyaluronic Acid (HA) in dentistry, I suggest this article: [Surface Treatment of the Dental Implant with Hyaluronic Acid: An Overview of Recent Data. Int J Environ Res Public Health. 2021 Apr 27;18(9):4670. doi: 10.3390/ijerph18094670.].
Authors’ Response: The paper has been cited.
- Reviewer’s comment: Line 232: Put a comma before “when”, because is a long sentence.
Authors’ Response: Comma has been added (line 340)
- Reviewer’s comment: Line 238: I think that the correct verb tense here is “were” and please improve the discussion of these two different biofilms.
Authors’ Response: The discussion of these two different biofilms has been improved with some edits and additional discussions (line 352-365)- We found that there was 24% more biomass in the biofilm formed by single cells at 103 CFU/mL than the biofilm by the preformed aggregates containing same cell concentration. However, for a higher inoculum concentration of 108 CFU/mL, this difference was only 4%. This could be due to the single cells grow more rapidly than the aggregates formed by low concentration of cells. On the other hand, the rate of growth of aggregates increases as the size of initial aggregates increases. So, we found a relatively small difference in case of aggregates from higher density cells. Similarly, the biofilm formed by single cells at 103 CFU/mL was 5% more thicker than the one formed by the aggregates with same cell concentration. Whereas, for the higher inoculum concentration (108 CFU/mL), the biofilm formed by single cells was 11% less thick than the biofilm formed with the aggregates. This could be potentially due to the fact that aggregates formed by cells at lower concentration are significantly smaller than the aggregates formed by cells at higher concentration. Therefore, biofilm formed by larger aggregates attached on the surface could be relatively thicker than the ones formed by the single cells or aggregates formed by lower concentration.
- Reviewer’s comment: Line 240: You wanted to say “inoculum” here or not?
Authors’ Response: We agreed and have now corrected this.
- Reviewer’s comment: Lines 249-251: Put SF.
Authors’ Response: SF has been added.
- Reviewer’s comment: Lines 254-263: This part is very confused and needs an improvement; the sentences are with two meanings and aren’t all of them so clear to understand your final discussion. You can add something as a future perspective or what is going to help the clinician or…”SO YOUR GOLD MESSAGE WHAT IS IT?”
Authors’ Response: The portion suggested by the reviewer has been revised (line 388-398)- Furthermore, we acknowledge some limitations of this study such as the characterization of biofilm after 24 hours of growth period when the biofilms were relatively saturated. We believe that initial events in terms of bacterial attachment initiated with lower and higher concentration inocula would provide more insights towards biofilm formation process. Thus, we aspire to conduct this study with shorter time periods to observe the initial events in the biofilm formation process. In addition, we also intend to see the effects of SF-mediated aggregates in the strength of biofilm formed over a longer period of 5 – 7 days. Therefore, with this study we presented evidence of the role of aggregates in biofilm formation, and we intend to do further research to characterize the early events in the biofilm formation process and long-term health of mature biofilm that were initiated with SF-mediated aggregates. Theses series of studies could provide new knowledge base on understanding the biofilm formation process from initiation to maturation in PJI.
- Reviewer’s comment: Conclusions
This section requires to be clearer; check the English as well!
Authors’ Response: The section has been modified to make it clearer.
- Reviewer’s comment: Lines 265-270: Here you have repeated your results and I suggest to reformulate this paragraph adding your final and real conclusion to be understandable for
Authors’ Response: The conclusion has been reformulated (line 409-414)- The obtained results provided evidence that S. aureus aggregated in presence of synovial fluid can form biofilm. The formation of biofilm thus follows a similar developmental time-period and antibiotic tolerance as more traditionally inoculated in vitro flow cell biofilms. While we acknowledge that future studies will be necessary to fully understand the extent of synovial fluid-initiated biofilm formation, our study provides evidence that S. aureus interaction with synovial fluid has an important role during biofilm formation and PJI.
Reviewer 2 Report
The properties of free-floating bacterial aggregates formed in synovial fluid and their relationship with bacterial biofilms formed on prosthetic surfaces represent a topic of great interest. Here, the authors report on the different characteristics of biofilms generated on titanium surfaces in vitro in a flow cell system, following the seeding with either pre-aggregated or single planktonic cells of Staphylococcus aureus.
Comments
1) In Materials and Methods, some relevant information is missing. No information is provided on the source of culture media and reagents. Was the synovial fluid human or bovine? What was the titanium grade of the coupons? Was the antibiotic solution for treating Ti coupons made in tryptic soy broth? What was the software used for the statistical analysis?
2) Figures 7 to 9 have not been introduced in the text of the manuscript. In some parts of subparagraph 3.3., the description of the results without explicitly referring to the corresponding figures generates some confusion.
3) Data illustrated in figure 9 should be preferably plotted in a logarithmic graph to better and more easily appreciate the descriptions provided in the text.
4) The authors speculate that residual synovial fluid in biofilms derived from pre-aggregated cells could be the reason for the increased susceptibility of these biofilms to the exposure to vancomycin. However, a generic antibacterial activity of synovial fluid should have affected the bacteria even in the absence of vancomycin. Conversely, the bacterial permeability protein was reported to be cytotoxic to Gram-negative bacteria and this is not the case as the tests were performed on S. aureus. Probably, this part could be slightly rephrased and better argued, describing even the limitations of the formulated hypotheses.
Minor points
1) Throughout the manuscript, there are numerous grammatical errors of subject-verb agreement that need correction (for instance, see row 11, row 28, row 54, row 55, row 73, row 77, row 96, row 100, row 103, and so on). However, the text should be carefully checked also for other types of mistakes.
2) Row 186: "... 2 and 1 log reductions of bacterial cells were ...".
3) In the captions of the figures concerning confocal microscopy a detailed explanation should be provided for the merged images (which stains were merged?) and for the merged GFP-PI images.
4) In the caption of figure 9, please, provide an explanation on the different level of statistical significance for *, **, and ***.
5) Figure 4: surface area Y-axis values should be all in the same format.
Author Response
Responses to reviewer 2 comments
The properties of free-floating bacterial aggregates formed in synovial fluid and their relationship with bacterial biofilms formed on prosthetic surfaces represent a topic of great interest. Here, the authors report on the different characteristics of biofilms generated on titanium surfaces in vitro in a flow cell system, following the seeding with either pre-aggregated or single planktonic cells of Staphylococcus aureus.
- Reviewer’s comment: In Materials and Methods, some relevant information is missing. No information is provided on the source of culture media and reagents. Was the synovial fluid human or bovine? What was the titanium grade of the coupons? Was the antibiotic solution for treating Ti coupons made in tryptic soy broth? What was the software used for the statistical analysis?
Authors’ Response: All the relevant information suggested by the reviewer has been added in the manuscript. For titanium, additional details have been added (line 128-130)
- Reviewer’s comment: Figures 7 to 9 have not been introduced in the text of the manuscript. In some parts of subparagraph 3.3., the description of the results without explicitly referring to the corresponding figures generates some confusion.
Authors’ Response: Figures 7-9 has been introduced in the manuscript text. Also, the results have been explained with corresponding figures.
- Reviewer’s comment: Data illustrated in figure 9 should be preferably plotted in a logarithmic graph to better and more easily appreciate the descriptions provided in the text.
Authors’ Response: Figure 9 has been plotted in log scale with increased font size.
- Reviewer’s comment: The authors speculate that residual synovial fluid in biofilms derived from pre-aggregated cells could be the reason for the increased susceptibility of these biofilms to the exposure to vancomycin. However, a generic antibacterial activity of synovial fluid should have affected the bacteria even in the absence of vancomycin. Conversely, the bacterial permeability protein was reported to be cytotoxic to Gram-negative bacteria and this is not the case as the tests were performed on S. aureus. Probably, this part could be slightly rephrased and better argued, describing even the limitations of the formulated hypotheses.
Authors’ Response: Additional information regarding SF antibacterial activity has been added in the manuscript (line 382-387)- In case of Gram-positive bacteria, the proteins such as lysozyme, lactoferrin, and secretory phospholipase might have played role in the bactericidal activity of SF observed against the Staphylococcus and Streptococcus species as speculated by other study. These proteins are believed to exists in the SF of healthy, traumatized, or osteoarthritis knees and provide innate defense against gram-positive organisms.
- Reviewer’s comment: Throughout the manuscript, there are numerous grammatical errors of subject-verb agreement that need correction (for instance, see row 11, row 28, row 54, row 55, row 73, row 77, row 96, row 100, row 103, and so on). However, the text should be carefully checked also for other types of mistakes.
Authors’ Response: Thorough review of the manuscript has been completed with suitable corrections.
- Reviewer’s comment: Row 186: "... 2 and 1 log reductions of bacterial cells were ...".
Authors’ Response: The error has been corrected.
- Reviewer’s comment: In the captions of the figures concerning confocal microscopy a detailed explanation should be provided for the merged images (which stains were merged?) and for the merged GFP-PI images.
Authors’ Response: Detailed explanation has been added in the figure captions of all confocal images.
- Reviewer’s comment: In the caption of figure 9, please, provide an explanation on the different level of statistical significance for *, **, and ***.
Authors’ Response: Explanation on the different level of statistical significance has been added in the figure 9 legend.
- Reviewer’s comment: Figure 4: surface area Y-axis values should be all in the same format.
Authors’ Response: It has been corrected.
Reviewer 3 Report
Here I present the review of the paper entitled “Free-floating aggregate and single cell initiated biofilms of 2 Staphylococcus aureus” submitted to Antibiotics.
Study describes bacterial formation on titanium surfaces. Papers is written clearly and brings novel data. However, some imperfection must be corrected before publication.
Critical issues
- There is no information whether bacterial strain was isolated form patient or reference ones.
- Discussion is poorly written and cannot be accepted in present form. Authors do not compare their data with sufficient amount of literature.
Major issues
- In the introduction what is biofilm should be described.
- Please provide justification for chosen vancomycin concentration.
- Languages and spelling check is required.
Minor issues
- In the title dash is missing between “cell” and “Initiated”.
Comments
Below comments are not necessary criticism of your work, please consider them as possibilities in future work.
- Usage of only one bacterial strain is strongly limiting factor for data extrapolation.
- Sometimes it is beneficial to use both reference and patient-isolated bacterial strains.
With kind regards,
Reviewer
Author Response
Response to reviewer 3 comments
- Reviewer’s comment: Here I present the review of the paper entitled “Free-floating aggregate and single cell initiated biofilms of 2 Staphylococcus aureus” submitted to Antibiotics.
Study describes bacterial formation on titanium surfaces. Papers is written clearly and brings novel data. However, some imperfection must be corrected before publication.
Critical issues
There is no information whether bacterial strain was isolated form patient or reference ones.
Discussion is poorly written and cannot be accepted in present form. Authors do not compare their data with sufficient amount of literature.
Authors’ Response: Citation has been provided for the bacterial strain used in this study.
The discussion section has been improved with additional references and edits (Lines 329-330,344-345, 353-366,384-401).
Major issues
- Reviewer’s comment: In the introduction what is biofilm should be described.
Please provide justification for chosen vancomycin concentration.
Languages and spelling check is required.
Authors’ Response: Biofilm description has been added in the introduction (line 56-59). The justification for chosen vancomycin concentration has been added in the discussion section (line 373-374).
Languages and spelling checks have been conducted for the entire manuscript.
Minor issues
- Reviewer’s comment: In the title dash is missing between “cell” and “Initiated”.
Authors’ Response: Dash has been included between cell and initiated.
Comments
- Reviewer’s comment: Below comments are not necessary criticism of your work, please consider them as possibilities in future work.
Usage of only one bacterial strain is strongly limiting factor for data extrapolation.
Sometimes it is beneficial to use both reference and patient-isolated bacterial strains.
Authors’ Response: We appreciate the reviewer’s recommendations and will consider implementing them in future experiments. The authors put it in discussion as well (line 398-400) - In future we also intend to use reference and patient-isolated bacterial strains to further explore this study as usage of a single bacterial strain might be a limiting factor for conclusive results.
Round 2
Reviewer 3 Report
I do not have more critical comments.
Kind regards